# Transformers significantly improve splice site prediction
Benedikt A. Jónsson [1,2], Gísli H. Halldórsson [1], Steinþór Árdal[1,2], Sölvi Rögnvaldsson[1], Eyþór Einarsson [1], Patrick Sulem [1], Daníel F. Guðbjartsson [1,2], Páll Melsted[1,2], Kári Stefánsson [1,2] ✉ & Magnús Ö. Úlfarsson [1,2] ✉

Mutations that affect RNA splicing significantly impact human diversity and disease. Here we present a method using transformers, a type of machine learning model, to detect splicing from raw 45,000-nucleotide sequences. We generate embeddings with residual neural networks and apply hard attention to select splice site candidates, enabling efficient training on long sequences. Our method surpasses the leading tool, SpliceAI, in detecting splice sites in GENCODE and ENSEMBL annotations. Using extensive RNA sequencing data from an Icelandic cohort of 17,848 individuals and the Genotype-Tissue Expression (GTEx) project, our method demonstrates superior performance in detecting splice junctions compared to SpliceAI-10k (PR-AUC = 0.834 vs. PR-AUC = 0.820) and is more effective at identifying disease-related splice variants in ClinVar (PR-AUC = 0.997 vs. PR-AUC = 0.996). These advancements hold promise for improving genetic research and clinical diagnostics, potentially leading to better understanding and treatment of splicing-related diseases.

RNA splicing is a major source of genetic diversity and a primary link between genetic variation and disease[1,2]. Although the biochemical mechanism of splicing is fairly well understood it remains difficult to identify splice variants outside the essential GT and AG dinucleotides positioned at exon-intron boundaries[3,4]. Pathogenic variants in non-coding regions play a significant role in rare genetic diseases, however, they have largely been overlooked in clinical practice due to difficulties in determining their impact[5]. Such variants can be detected from biological samples using functional assays[6] and RNA-Sequencing (RNA-Seq)[7]. Splicing quantitative trait loci (sQTLs) are a common class of variants that are associated with the usage of alternative splice sites in RNA and can be detected using RNA-Seq. However, developing computational methods that can accurately detect such variants would be more cost-effective, since they do not require the collection and processing of biological samples. Such methods are therefore well suited for large-scale analysis and variant prioritization. Recently, splice site prediction methods, such as SpliceAI[8], have been shown to be effective at detecting splice site variants. SpliceAI is a deep neural network, based on a convolutional neural network (CNN) architecture and it has been shown to outperform other methods[8,9] and to improve pathogenicity predictions of sequence variants[10]. The detection of sequence variants causing aberrant splicing can further be improved by integrating the SpliceAI predictions with RNA-Seq data[11] and by training the SpliceAI architecture on collapsed versions of transcripts[12]. The latter is in line with the observation that accounting for long-range sequence determinants improves splice

site predictions[8]. While CNNs can learn long-range dependencies by using large convolution kernels or by using multiple convolutional layers and pooling operations, they are not efficient at modeling long-range dependencies[13]. Other architectures, such as transformers, are in principle better suited for learning long-range dependencies, since they allow each input element to interact with every other input element, enabling transformers to capture complex relationships between distant elements[14]. Transformers have been highly successful in natural language processing[15] and other diverse tasks such as image recognition, protein folding, prediction of gene expression, and recently in human genomics[16–19]. Methods such as the Nucleotide Transformer[19] and DNABERT-2[20] can perform accurate splice site prediction, however, currently, it is difficult to scale them to long sequence context. This is because self-attention scales quadratically with sequence length.

Allowing transformers to scale well with long contexts is an active research area[21–26]. However, learning from input sequences longer than 10,000 elements remains challenging[27]. Recently, alternative architectures such as RetNet[28] and Hyena[29] have been proposed to handle larger than 10,000 element contexts. However, they are either untested at splice site prediction or have not been demonstrated to be effective[30]. Another method used to increase the context size in transformers is to aggregate the input sequence by chunking it into k-mers or using tokenizers[19,20]. However, single nucleotide variations can have a large impact on splicing, and by aggregating, information on the single nucleotide level is lost[30].

---

[1]deCODE Genetics/Amgen Inc., Reykjavik, Iceland. [2]University of Iceland, Reykjavik, Iceland. ✉e-mail: kstefans@decode.is; mou@hi.is

We hypothesize that large contexts are essential for tracking other splice sites in the pre-mRNA sequence and that the usage of these splice sites depends on the simultaneous utilization of other splice sites. Based on these assumptions, we propose using hard attention[31,32] to select a set of candidate acceptors and donors splice sites before passing them to a transformer. Hard attention selects a single or a few discrete elements from the input sequence by making a binary decision on which parts of the input to attend to while soft attention, which is used in transformers, computes a weighted average over all elements of the input sequence. Using hard attention allows us to shorten the sequences that the transformer receives and to reduce the memory required to train the transformer (see "Model architecture and loss function" section for more details). In our experiments, we show that a transformer equipped with the proposed hard attention module and trained to utilize 45 knt sequence contexts, predicts splice sites with higher accuracy than SpliceAI-10k. We show *CREBRF* as an example of a gene where the transformer model detects a splice site that SpliceAI-10k fails to detect. Additionally, we validate our method on RNA-Seq data from a large Icelandic cohort of RNA sequenced blood samples collected from 17,848 individuals and GTEx V8[33], which consists of 15,201 RNA-Seq samples from 49 tissues and 838 individuals, and demonstrate that it has greater accuracy than SpliceAI-10k detecting splice junctions not present in ENSEMBL v87 and essential splice site sequence variants in the Icelandic cohort, in addition to detecting pathogenic splice variants from ClinVar[34] with higher accuracy than SpliceAI.

## Results

### Transformers improve splice site detection

We compared the performance of our model, Transformer-45k, with SpliceAI-10k on splice site annotations from ENSEMBL transcripts[35], held out during training. The method was tested against both SpliceAI-10k trained on GENCODE annotations that is similar to the training set from the SpliceAI paper, and a version of SpliceAI-10k trained on the ENSEMBL training set. Additionally, we trained a model called Transformer-10k which has the same architecture as Transformer-45k and was trained and evaluated on the same 10kb sequence context as SpliceAI. The models were trained on transcripts labeled as protein-coding and were also tested on GENCODE transcripts held out during training. Overall, Transformer-45k has better performance than SpliceAI-10k with regard to the metrics we tested. It has higher PR-AUC and top-k accuracy than SpliceAI-10k on both datasets (Fig. 1b). Looking specifically at the decision thresholds used for top-k accuracy calculations (Transformer-45k [acceptor: 0.446, donor: 0.485], SpliceAI-10k [acceptor: 0.460, donor: 0.507]), we see that Transformer-45k detects 744 more splice sites (out of 179,424).

To get a clearer picture of the difference between the models, we decided to look specifically at sites where the predictions disagree. We measure disagreement between the models by calculating the total variation distance (TVD)[36] between predictions and we label predictions to be in disagreement if the TVD is higher than 0.1. In the held-out ENSEMBL set, the models disagree on about 1 in 20,000 predictions, i.e., 32,673 out of

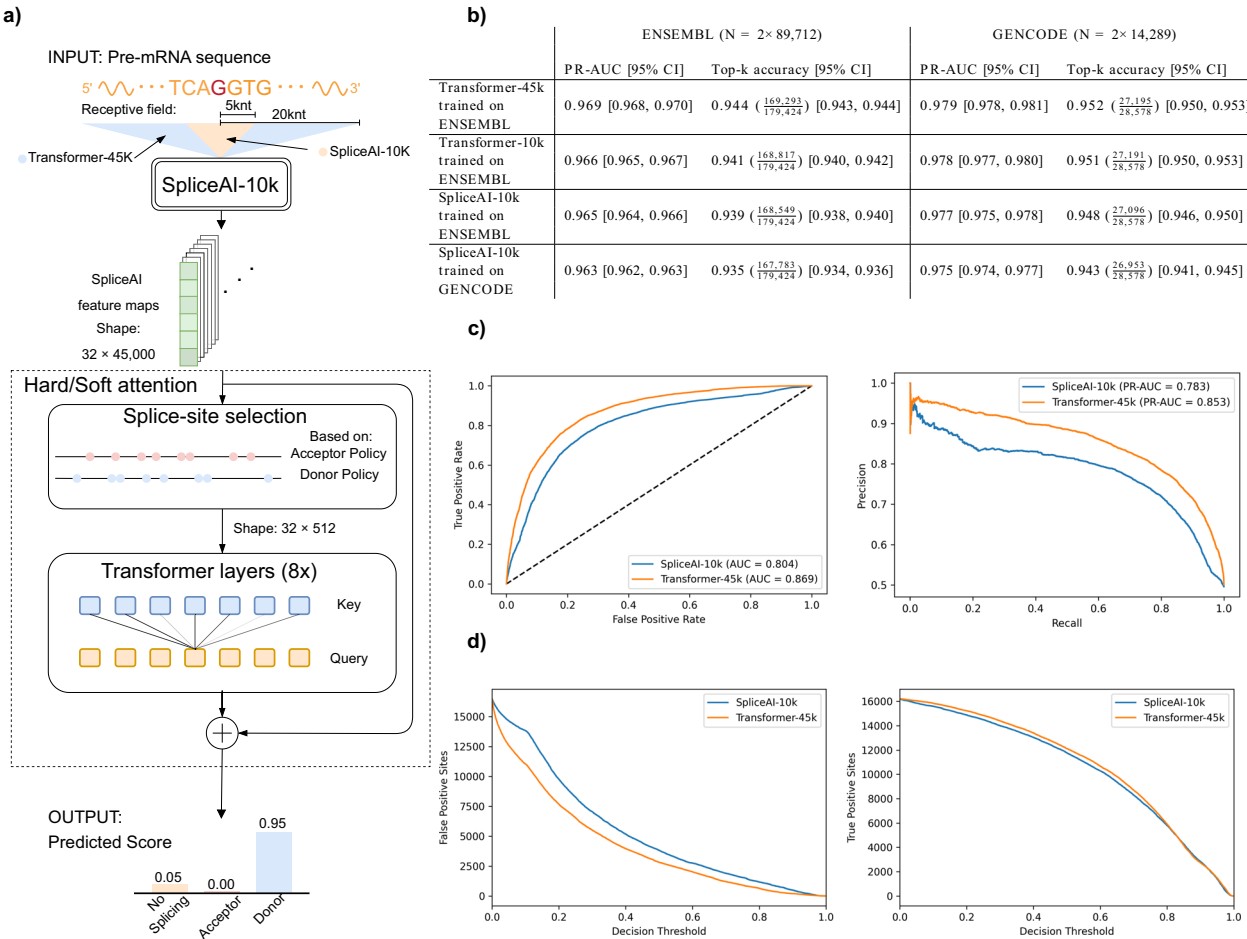

**Fig. 1 | A diagram showing Transformer-45k and a performance comparison with Splice-10k. a** For each position in an input DNA sequence, the method looks at the surrounding context region and outputs a predicted score for three options: no splicing, acceptor, or donor. **b** Comparison of Transformer-45k with SpliceAI-10k on both ENSEMBL and GENCODE annotations with regard to area under the precision-recall curve (PR-AUC) and top-k accuracy. 95% confidence intervals (CIs) are shown in brackets. *N* denotes the number of splice sites in the test set, not the total size. E.g., the total size of the ENSEMBL test set is 664,940,000 nt. **c** Receiver operating characteristic (ROC) curve and precision-recall curve for cases where SpliceAI and Transformer-45k disagree (TVD ≥0.1). **d** The total number of false positive and true positive splice sites as a function of the decision threshold for cases where SpliceAI and Transformer-45k disagree (TVD ≥0.1).

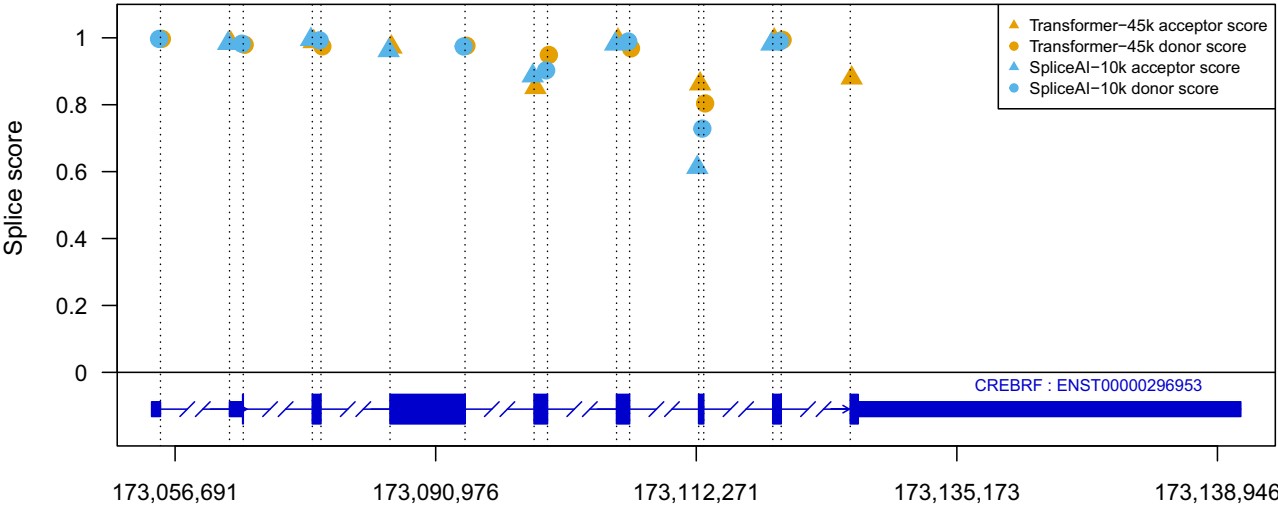

**Fig. 2 | A comparison of Transformer-45k and SpliceAI-10k splice site predictions for *CREBRF*.** The predictions are mostly in agreement, except SpliceAI-10k does not detect the acceptor for the final exon.

664,940,000 predictions. For this subset, we observe that Transformer-45k is more likely to predict the splice sites correctly. Here, Transformer-45k has a 0.869 ROC-AUC and 0.853 PR-AUC, compared to 0.804 ROC-AUC and 0.783 PR-AUC for SpliceAI-10k (Fig. 1c). For these locations, the observed boost in performance seems to be mainly due to Transformer-45k making fewer false positive splice site predictions because the models make about as many true positive predictions (Fig. 1d). We can also look at the top-k decision thresholds to look at the agreement between predicted splice sites, here we see that the models agree on 175,825 splice sites and 664,757,322 non-splice sites, and they disagree on 3599 splice sites and 3254 non-splice sites. For cases where the the predictions disagree, Transformer-45k has 0.609 accuracy (4172 correct sites) and SpliceAI-10k has 0.391 accuracy (2681 correct sites).

There is a high agreement between splice site predictions made by the two models, therefore it is important to highlight that small differences in splice site predictions can have a large impact on downstream analysis. For example, when predicting splice sites in *CREBRF*, Transformer-45k, and SpliceAI-10k only disagree on one splice site. However, in this case, SpliceAI completely misses the final exon of the gene (Fig. 2). In an attempt to understand how Transformer-45k improves the predictions, we looked at attention scores in Transformer-45k for known splice sites and observed that when making predictions for sites that are labeled as splice sites it primarily attends to other annotated splice sites in the same gene. E.g., in *CFTR* other known splice sites have the highest attention scores (Fig. S4).

## Fine-tuning on RNA-Seq data

To examine how well the predicted splice site agreed with splice sites observed in RNA-Seq data, we looked at RNA-Seq data from a large Icelandic cohort gathered by deCODE genetics and data from GTEx V8[33]. The Icelandic cohort consists of RNA-Seq samples from 17,848 individuals, collected from blood, and GTEx V8 is comprised of 15,201 RNA-Seq samples collected from 49 tissues of 838 individuals. By combining data from these two sources we were able to construct splice site annotations for 17,239 protein-coding genes, consisting of 360,601 acceptors and 359,934 donors in total. To accurately predict these splice site annotations, we found that it was necessary to fine-tune the models on the RNA-Seq splice site annotations. Here we split the RNA-Seq splice site annotations into training and test set by chromosome in the same fashion as in the previous section and fine-tuned models for four epochs. We found that with fine-tuning, Transformer-45k has higher splice site prediction accuracy than SpliceAI-10k on the held-out set (Table 1). Additionally, we fine-tuned the models

using the same procedure exclusively on GTEx RNA-Seq splice sites and observed that Transformer-45k continued to outperform SpliceAI-10k (Table 2).

Next, using the models fine-tuned on both GTEx and the Icelandic cohort, we tested their ability to detect unannotated splice junctions, i.e., splice junctions not present in the ENSEMBL v87 database, and sQTLs in the Icelandic RNA-Seq data and compared there performance to that of SpliceAI-10k with pre-trained weights. In the Icelandic cohort, there were 351,546 splice sites detected from RNA-Seq spliced alignments, 160,591 (45.7%) of which were not in an annotated transcript in the ENSEMBL v87 database. Using 0.1 as the decision threshold, to ensure high recall (Table S2), our method detects 98.5% of junctions annotated in ENSEMBL and 71.8% of unannotated junctions, while SpliceAI-10k with pre-trained weights detects 96.6% of the annotated junctions and 53.8% of unannotated junctions. In comparison, Transformer-45k fine-tuned only on GTEx splice site annotations detects 98.1% of junctions annotated in ENSEMBL and 67.6% of unannotated junctions.

## Table 1 | Transformer-45k and SpliceAI-10k performance on splice junctions from all tissues in GTEx V8 and Icelandic blood samples

|  | PR-AUC [95% CI] | Top-k accuracy [95% CI] |
|---|---|---|
| Transformer-45k fine-tuned on RNA-Seq annotations | **0.834** [0.833, 0.835] | **0.744** $(\frac{147,949}{198,984})$ [0.742, 0.745] |
| SpliceAI-10k fine-tuned on RNA-Seq annotations | 0.832 [0.830, 0.833] | 0.741 $(\frac{147,400}{198,984})$ [0.739, 0.742] |
| SpliceAI-10k pre-trained weights | 0.820 [0.819, 0.821] | 0.732 $(\frac{145,666}{198,984})$ [0.731, 0.734] |
| SpliceAI-10k trained on ENSEMBL | 0.753 [0.751, 0.754] | 0.686 $(\frac{136,550}{198,984})$ [0.685, 0.688] |
| Transformer-45k trained on ENSEMBL | 0.750 [0.749, 0.752] | 0.691 $(\frac{137,595}{198,984})$ [0.690, 0.693] |

The fine-tuning was done on the combined RNA-Seq splice sites, however, the results are only shown for chromosomes left out during training, here the combined number of splice sites is 198,984. The performance metrics are PR-AUC and top-k accuracy. 95% confidence intervals (CIs) are shown in brackets and the best score displayed in bold.

Again using the fined-tuned Transformer-45k, we compute delta scores from splice site predictions to quantify the effect of variants on splicing. Filtering the sQTLs based on the distance to the closest annotated splice site reveals that the accuracy of the delta score is highly dependent on the proximity of sQTLs to splice sites (Fig. 3a). We also looked at classification performance for classifying splice-disrupting and splice-creating variants, these are variants that are observed to change GT/AG splice motifs,

### Table 2 | Transformer-45k and SpliceAI-10k performance on GTEx V8 splice junctions

| | PR-AUC [95% CI] | Top-k accuracy [95% CI] |
|---|---|---|
| Transformer-45k fine-tuned on GTEx V8 | **0.849** [0.848, 0.851] | **0.758** ($\frac{137,139}{180,872}$) [0.757, 0.760] |
| SpliceAI-10k fine-tuned on GTEx V8 | 0.842 [0.841, 0.844] | 0.751 ($\frac{135,829}{180,872}$) [0.750, 0.753] |
| SpliceAI-10k pre-trained weights | 0.836 [0.835, 0.838] | 0.747 ($\frac{135,112}{180,872}$) [0.746, 0.749] |
| SpliceAI-10k trained on ENSEMBL | 0.775 [0.773, 0.777] | 0.706 ($\frac{127,737}{180,872}$) [0.705 0.709] |
| Transformer-45k trained on ENSEMBL | 0.772 [0.770, 0.774] | 0.711 ($\frac{128,651}{180,872}$) [0.710, 0.713] |

These results are only for chromosomes left out during training, here the combined number of splice sites is 180,872. The performance metrics are PR-AUC and top-k accuracy. 95% confidence intervals (CIs) are shown in brackets and the best score is displayed in bold.

and are therefore highly likely to truly affect splicing. Here, our method has 0.991 PR-AUC detecting essential splice site variants, where SpliceAI-10k has 0.981 PR-AUC (Fig. 3b). Selecting 0.1 as a decision threshold, our method detects 88.604% (out of 351) of splice-disrupting variants, and 88.698% of splice-creating variants (out of 699), where SpliceAI-10k detects 81.148% of splice-disrupting variants and 79.828% of splice-creating variants.

We evaluated the classification performance of the Transformer-45k model on 35,464 pathogenic splice variants from ClinVar. The Transformer-45k achieved a PR-AUC of 0.997, compared to SpliceAI's PR-AUC of 0.996 (Fig. 3c). To further analyze the predictive performance, we plotted a scatter plot of delta scores for these variants. The results indicate that pathogenic splice variants predominantly have delta scores around one, signifying high confidence in predictions. Non-splicing variants, on the other hand, cluster around a delta score of zero, indicating a low likelihood of splicing disruption. Interestingly, benign splice variants, though fewer in number ($n = 1001$), exhibit a wider distribution of delta scores, which are less consistently clustered around zero and one (Fig. 3d). This variability suggests a more complex prediction landscape for benign splice variants.

## Discussion

In this study, we have looked at predicting splice sites with transformers and shown that they can learn to utilize long sequence contexts to predict splicing with better classification accuracy than the current best splice site prediction methods in the literature. We tested our method on splice site annotations from ENSEMBL and GENCODE and showed that it was able to predict splicing with greater accuracy than SpliceAI-10k, both with regard

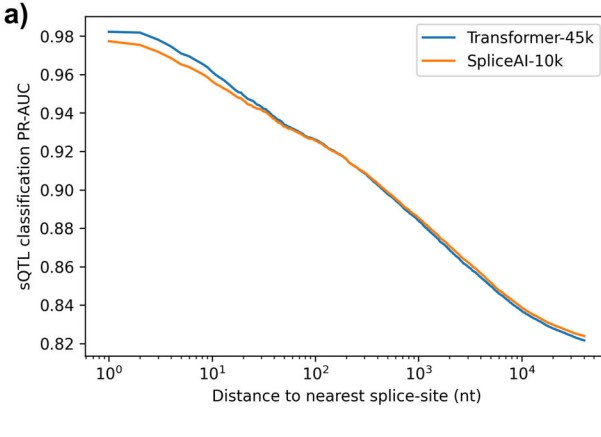

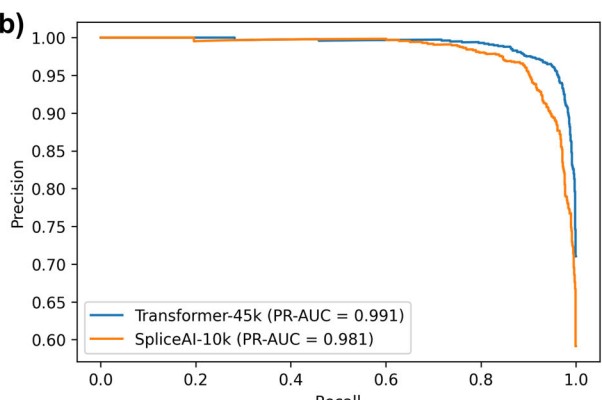

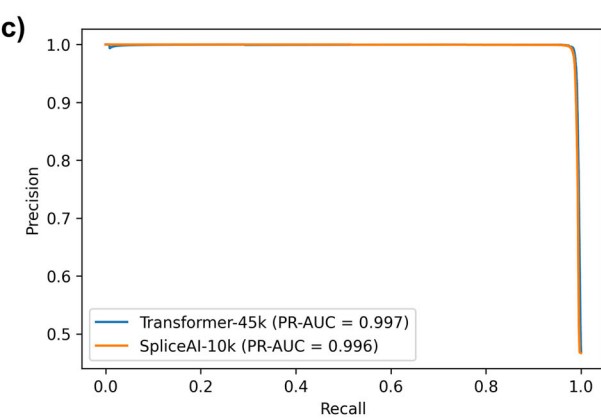

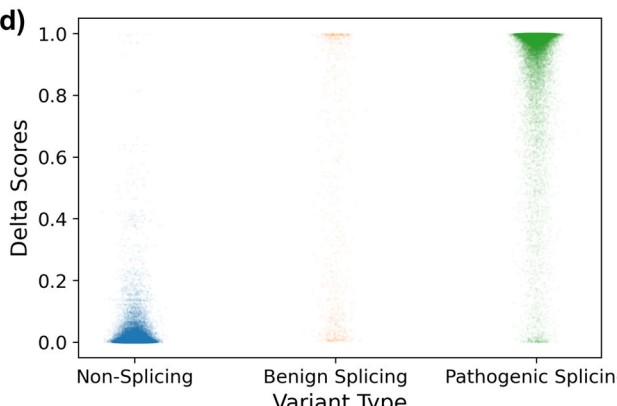

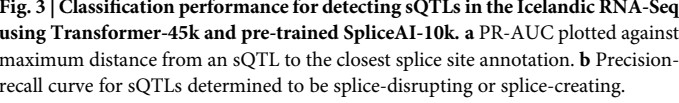

**Fig. 3 | Classification performance for detecting sQTLs in the Icelandic RNA-Seq using Transformer-45k and pre-trained SpliceAI-10k. a** PR-AUC plotted against maximum distance from an sQTL to the closest splice site annotation. **b** Precision-recall curve for sQTLs determined to be splice-disrupting or splice-creating.

**c** Precision-recall curve for 35,464 pathogenic splice variants in ClinVar. **d** A scatter plot showing the distribution of delta scores for non-splicing variants ($n = 40,528$), benign splice variants ($n = 1001$), and pathogenic splice variants ($n = 35,464$).

to PR-AUC and top-k accuracy. Focusing on the splice site predictions where our method disagreed with SpliceAI-10k, we saw that our method makes fewer false positive predictions while making about as many true positive predictions. By providing the transformer with a list of 512 potential splice sites, we enable it to produce more accurate predictions than those achieved with SpliceAI alone. This improvement may be attributed to the model's ability to learn the dependencies between splice sites over a larger sequence context, supporting the hypothesis that longer raw sequences are beneficial for capturing splice site interactions.

When classifying unannotated splice junctions and splice variants, we found that fine-tuning the model on RNA-Seq data was necessary to achieve better performance than SpliceAI-10k. This is likely due to our training set only consisting of protein-coding ENSEMBL transcripts. Genes can have multiple transcript annotations and splice sites observed in RNA-Seq can come from any one of these transcripts. The ENSEMBL annotations can be combined into a single gene annotation and this could improve performance for detecting splice sites in RNA-Seq data. However, this would result in less training data, and many splice sites observed in RNA-Seq would still be missing from the annotations.

Since we used splice site delta scores to classify sQTLs, it is important to note that sQTLs are associations between variations in splicing and sequence variants and are therefore not necessarily caused by splice site mutations. We observed that sQTLs are likelier to have a high splice delta score if they are close to annotated splice sites. This could be due to sQTLs with proximity to annotated splice sites being more likely to be caused by splice site mutations and this makes it harder to interpret the classification accuracy results. Limiting the sQTLs to sequence variants in essential splice sites helps filter the set down to sequence variants that are highly likely to be splice mutations. However, it should be noted that it is possible to find these sequence variants based on splice site annotations and splice site prediction methods are usually not necessary for detecting them. Even in these cases, accurate splice site prediction is of interest, since current methods cannot reliably predict whether exon-skipping, cryptic activation, or multiple events will result from these mutations and this complicates clinical interpretation of pathogenicity[37].

We conducted an additional experiment in Fig. 1b to determine whether the observed performance improvement can be attributable to the transformer architecture or the increased sequence context. We trained a transformer model using a 10kb context and compared its performance to both SpliceAI and our 45kb context transformer model. The 10kb context transformer model outperformed SpliceAI, confirming that the transformer architecture contributes to more accurate predictions. Nevertheless, the 45kb context transformer model achieved the highest performance, highlighting that an extended sequence context is a significant factor in improving model accuracy. In principle, our method can be trained on larger contexts than 45,000 nt, and there is no reason to assume that increasing the context further will not be beneficial. The same applies to the number of selected splice sites and parameters such as depth and number of heads. However, these models may need more training data or longer training to show any improvements over our current method.

A challenge with constructing splice site annotations from RNA-Seq data is that the quality of the annotations is likely not as high as if only annotations from ENSEMBL were used. Some of the splice sites detected in RNA-Seq can be due to sequence variants and it is not possible to predict these splice sites without including information about the sequence variants. On the other hand, it has also been shown that augmenting splice site annotations with RNA splicing measurements of multiple different species can allow more accurate prediction of splicing in different tissue types[38].

A known issue with policy gradient methods is their tendency to exhibit high gradient variance, this can slow down convergence or prevent the model from reaching optimal policies. Despite this, our model quickly learned a policy that selected almost all annotated splice sites. However, reducing gradient variance could potentially further refine the policy and improve model performance.

In conclusion, we designed a splice site prediction method that utilizes transformers and showed that they can significantly improve the state-of-the-art. Our method utilizes hard attention to reduce pre-mRNA sequences to a set of potential splice sites, that have a manageable length for transformers to learn long-range dependencies between splice sites. The model is trained on a about four times larger context than SpliceAI and Nucleotide Transformer v2. In our experiments, we showed that the Transformer-45k makes fewer false positive predictions than SpliceAI while predicting about as many true positives. We observed, that Transformer-45k primarily attends to other annotated splice sites when performing splice site predictions. Finally, when our method is fine-tuned on RNA-Seq data from a large Icelandic cohort and GTEx V8, it detects more unannotated splice junctions, essential splice variants, and pathogenic splice variants than SpliceAI.

## Methods

### Preprocessing

The splice site annotations we used are based on data from ENSEMBL V87[35] and GENCODE V39[39]. Only protein-coding transcripts with one or more splice junctions were used and transcripts on chromosomes 1, 3, 5, 7, and 9 were held out for the test set. In ENSEMBL we only selected transcripts with support level 1. This resulted in an ENSEMBL training set that has 22,375 transcripts and a GENCODE training set with 13,384 transcripts. The corresponding ENSEMBL test set includes 8955 transcripts and the GENCODE test set includes 1652. Before training we removed 10% of transcripts from the training set and placed them in a validation set, in the ENSEMBL-based annotations 21,432 splice junctions were selected into this set. Nucleotides were one-hot encoded as as A = [1,0,0,0], C = [0,1,0,0], G = [0,0,1,0], and T = [0,0,0,1]. The labels were encoded as 'no splicing' = [1,0,0], 'acceptor' = [0,1,0], and 'donor' = [0,0,1].

Nucleotide sequences are stored in sparse arrays split by chromosome, where nucleotides outside of genes are stored as zeros. The array indices correspond to nucleotide chromosome position and to deal with negative-strand genes we reverse complement the nucleotide sequences on the fly. This allows us to easily change the context and sequence length without needing to write a copy of the sequence to disk.

### Model architecture and loss function

The proposed method consists of three main parts, an encoding module, a selector module, and the transformer module. All three parts of the model are trained simultaneously from scratch and optimized with the following loss function:

$$\text{Loss} = \text{Cross-entropy loss} + \lambda \cdot \text{Policy loss}. \quad (1)$$

This combined loss function is designed to simultaneously quantify the models' proficiency at splice site classification (Cross-entropy loss) and the selector modules' ability to select relevant splice sites (Policy loss). The Policy loss is scaled by a factor $\lambda$ and during training, it is set to $10^{-6}$. To train the model we use a 2D cross-entropy loss:

$$\text{Cross-entropy loss} = \sum_{i=1}^{N} \sum_{j=1}^{3} y_{i,j} \log(p_{i,j}), \quad (2)$$

where $N$ is the length of the sequence context, $y_{ij}$ is a one-hot encoded splice site label ('no splice', 'acceptor', and 'donor'), and $p_{i,j}$ is a splice site prediction.

The encoder module takes a pre-mRNA sequence as input and maps each position in the sequence to a 32-dimensional (32D) feature space based on its context. Nucleotides in the input sequence are one-hot encoded and mapped using a CNN that has the same architecture as SpliceAI-10k (Fig. 1a). By doing this, the module can learn to encode information for each sequence position from its surrounding 10k context. We base the encoder architecture on SpliceAI since it has been thoroughly tested and shown to be effective at splice site prediction. This allows us to focus on designing other parts of the model.

The purpose of the selector module is to reduce the sequence length down to a manageable size for the transformer. This module receives an encoded sequence of nucleotides and uses hard attention to select 512 candidate splice sites out of the sequence (Fig. S1). The selector module uses a policy, parameterized by a feed-forward network, to fill 512 slots, that are split evenly between acceptors and donors, with as many annotated splice sites as possible. The policy is trained with a policy gradient loss[40], defined as:

$$\text{Policy loss} = -\frac{1}{S}\sum_{s=1}^{S}\sum_{t=1}^{512}\log\left(\pi_\theta\left(a_s^t|X_s, C_s^{t-1}\right)\right)R_s^t, \quad (3)$$

where the policy $\pi_\theta$ is the probability of taking action $a_s^t$ at step $t$ and trajectory $s$, given an embedding $X_s$ and previous actions $C_s^{t-1}$. $C_s^{t-1}$ is an indicator vector that masks out previous actions and prevents the policy from selecting the same splice site twice. Finally, $S$ is the total number of trajectories and $R_s^t$ is the reward. In practice, we found that using one trajectory for each sequence was enough to achieve stable training. We want the module to select the annotated splice sites and also select promising functional splice sites that are not in the annotations, therefore, annotated splice sites receive reward $R_s^t = 1$ and other sites $R_s^t = 0$. This ensures that the policy is not penalized for selecting non-splice sites. An exception is made if the acceptor selector selects an annotated donor, and vice versa, to discourage the selector from selecting the wrong splice type, here a $R_s^t = -1$ penalty is given. The policy is parameterized by a fully connected feed-forward network with 32D vector input, one hidden layer with four units and a leaky ReLU activation, and two outputs. The policy network learns to take embeddings from the encoder module as inputs and returns acceptor and donor site logits as outputs. During training these logits are used to parameterize two categorical distributions, for each splice site type, over the 45,000 nt context. The policy alternates between sampling acceptor and donor sites from the distributions, without replacement, until it has selected 512 potential splice sites. However, during test time, we simply select the acceptors and donors with the largest logits.

The transformer module consists of eight transformer encoders with four heads. The input to the transformer module is the 512 sites from the selector. These sites are embedded with the encoding module, as previously described, and a fixed sinusoidal position encoding is added to them. The transformer encoders consist of a gated multi-headed self-attention layer, a feed-forward layer, and skip connections between them. Inputs to the self-attention layer and feed-forward layer are pre-layer normalized with LayerNorm[41,42]. The gated multi-headed self-attention layer learns key, query, and value matrices, where the input has dimension $d = 32$, and uses a gated attention function[17] (Fig. S2), defined as:

$$\text{Gated attention}\,(Q, K, V, G) = \sigma(G)\text{softmax}\left(\frac{QK^T}{\sqrt{d}}\right)V, \quad (4)$$

where $K$, $Q$, and $V$ are key, query, and value matrices and $G$ is a 32D linear map that is calculated from the input embedding, and $\sigma$ is a sigmoid activation function. In addition, a fully connected feed-forward network is included after each attention layer. It has input and output dimensions equal to $d$ and has one hidden layer with dimension 512 and GELU activation[43].

The output of the transformer module is finally sent to the prediction head. This is a convolutional layer with kernel size one and a softmax activation function, it maps 32D feature maps down to three feature maps, where the three possible outputs correspond to 'no splice', 'acceptor', and 'donor'.

## Model training

All models were trained for 10 epochs with the AdamW optimizer[44] and with 96 samples per batch. The AdamW learning rate was set to 0.002, with $\beta_1 = 0.9$, $\beta_2 = 0.999$, $\epsilon = 1e{-}08$, and weight decay = 1e−05. We used linear warm-up for the first 1000 optimization steps. After five epochs the learning rate was reduced by half each epoch. The model weights were randomly initialized ten times and trained. Training the model for ten epochs with 3 NVIDIA A100 GPUs takes about 9 hours.

SpliceAI-10k was retrained on data and code made available by Jaganathan et al.[8]. The original model was implemented using Keras (version 2.0.5) with TensorFlow backend and is trained on a GENCODE annotations constructed by the authors. Additionally, we implemented the model using PyTorch and constructed a training set using ENSEMBL annotations. The reported results for the methods trained on ENSEMBL are the average predictions of ten models.

To fine-tune the models on data from the Icelandic RNA-Seq cohort and GTEx V8, weights from the ENSEMBL dataset training runs were used as a starting point and trained for four additional epochs on splice sites obtained from RNA-Seq. During fine-tuning all weights were kept trainable and the learning rate was set to 2e−4.

## Processing of RNA-Seq data

The RNA-Seq data from the Icelandic cohort consist of 17,848 samples drawn from blood, from the same number of individuals (9784 females, 8064 males) collected using Illumina NovaSeq and HiSeq machines with read length $2 \times 125$ and poly-A mRNA isolation. These samples were aligned separately to the maternal and paternal inherited genome references using STAR v2.5.3a. Subsequently, we transferred the alignment files (BAM) to GRCh38 reference space (updating CIGAR and POS fields), merged the two files into a single BAM file, and annotated the parental alignment with a higher alignment score as primary alignment. The alignment files were scanned to detect splice sites from the CIGAR strings of primary alignment. Alignment counts per splice site were gathered on the fragment level and annotated with information on multi-mapping and length of sequence overhang aligned to aside exons. Splice sites were included if one individual fulfilled the following splice count requirements; (1) at least 4 fragments mapped, (2) maximum of shorter overhang is larger than 7 base pairs, (3) log2 entropy of left and right overhang length is larger or equal to two and (4) donor or acceptor site is within annotated gene boundary. Using aggregated data from all individuals, splice sites were filtered out if multi-mapped alignment excited more than 20% of mapped alignments or if the maximum fragment count was less than 5% of the expected transcript abundance. After filtering 351,546 splice sites were used in subsequent analysis.

The filtered splice junction counts per individual were grouped into sets of overlapping splice junctions (SOSJ) with some shared acceptor or donor sites using an amended version of the LeafCutter algorithm, to handle the large dataset[45]. These sets of splice sites allowed us to quantify alternative splicing by calculating the percentage spliced in (PSI) per individual; the proportion of splice count divided by the total number of fragments aligned to any of the splice sites in the SOSJ. A cis-sQTL scan was carried out by testing for association between PSI and sequence variants closer than 30kb to annotated gene overlapping SOSJ. The most significant sequence variants associated with PSI were annotated as lead-sQTLs. The cohort was a homogeneous population of 17,848 Icelanders (9784 females, 8064 males). The year of birth (YOB) data was binned into 5-year intervals, with the oldest participants born closest to 1920 and the youngest born closest to 2005. The median YOB for both sexes was 1960. In the cis-sQTL association scan, we adjusted for both technical covariates and kinship, since the pedigree of Icelanders was available. Prior to the cis-sQTL analysis, PSI values were adjusted for technical covariates (median coverage variation, mapping rates, strand-specificity), RNA quality metrics (RIN value, concentration), sample characteristics (storage time, blood neutrophil percentage), and sequencing batch effects. Sex, determined from genotype data, and age was evaluated as a potential covariate but excluded due to minimal contribution to PSI variation. In total, we detected 257,372 lead-sQTL of which 146,372 are within genes and pass a basic quality filter (REF ≠ ALT). We detect 80,976 lead-sQTLs with $p$-values below the Bonferroni threshold ($\frac{0.05}{146,372}$). And of these, 1588 sQTLs disrupt highly conservative splice motifs GT/AG while 2113 sQTL, create the canonical splice motif. These variants are highly

likely to truly affect splicing and we refer to them as splice-disrupting if they remove a splice motif and splice-creating if they create a splice motif. Additionally, for each lead-sQTL, we constructed a list of variants in the vicinity of the lead-sQTL that RNA sequencing never detects to affect splicing. To assess the ROC of the delta scores, we randomly select one of these negative examples for each lead-sQTL.

For GTEx V8, junction counts were calculated using STAR v2.5.3a sequence alignments provided by GTEx. Splice junctions were filtered out if the junction did not have reads in four or more subjects, if either end of the junction is in an ENCODE blacklist region or a simple repeat region, and if the junctions were not a part of a LeafCutter cluster that includes canonical splice sites[45].

The splice site annotations used for fine-tuning our model were constructed by combining RNA-Seq splice junctions detected in all 49 tissues in GTEx and the Icelandic blood samples. Junction reads were selected if they were present in four or more individuals, not in blacklisted sequencing regions, and if either end of the junction was present in the canonical transcript for a gene. The combined set of splice site annotations consists of 360,601 acceptors and 359,934 donors from 17,239 genes. In comparison, using the same method to construct annotations using exclusively reads from tissues in GTEx V8, we identified 310,532 acceptors and 311,499 donors from 16,308 genes.

## Statistics and reproducibility

The occurrence of splice sites is rare compared to non-splice sites. Therefore, the performance metrics we choose need to be robust to unbalanced data. The metrics we used to evaluate the model performance were area under the precision-recall curve (PR-AUC) and top-k accuracy. Here we define the top-k accuracy in the same way as Jaganathan et al.[8]. That is, as the fraction of $k$ positions that are correctly predicted to belong to a class, where $k$ is the number of positions truly belonging to the class and the decision threshold is chosen so that exactly $k$ positions are predicted for this class. To calculate 95% confidence intervals for PR-AUC and top-k accuracy we performed bootstrapping with 1000 samples.

To find sites with disagreeing model scores, we calculated the TVD between predictions and selected predictions with a distance higher than 0.1. Since the predictions are probability distributions with a countable sample space the TVD simplifies to[36]:

$$\delta(P, Q) = \frac{1}{2} \sum_{\omega \in \Omega} |P(\{\omega\}) - Q(\{\omega\})|.$$

where $P$ and $Q$ are probability distributions, and $\Omega = \{$no splice, acceptor, donor$\}$.

To inspect attention patterns, we visualized the attention in transformer encoders by calculating, for a given input, the average value of all attention matrices, in all ten transformer-45k models.

Statistical analyses were conducted to identify and replicate splicing quantitative trait loci (sQTLs) in our cohort compared to those reported in the GTEx V8 whole blood dataset. In the GTEx analysis, significant sQTLs were determined using a false discovery rate (FDR) threshold of 5% ($q$-value < 0.05) to control for multiple testing. Replicates were defined as the lead-sQTLs identified in GTEx that were also present in our dataset. We assessed replication by testing these variants for association with the corresponding splicing events in our cohort. A replication was considered successful if the variant showed a significant association at a Bonferroni-adjusted $p$-value threshold $(\frac{0.05}{1,972})$. The majority (94.2% [1858 out of 1972]) of lead-sQTLs from GTEx were replicated in our cohort, indicating high reproducibility of the findings.

## Delta score

To compute the delta score we followed the procedure outlined by Jaganathan et al.[8]. We first calculate the difference between the predictions for an alternative sequence that includes a sequence variant and the prediction for the reference sequence. Then the location and splice site with the highest absolute difference in either the acceptor or donor site predictions is located. This difference is defined as the delta score and if the score is sufficiently high, it indicates a splice site gain or loss at that location.

## ClinVar variants

We downloaded the ClinVar variants in variant call format and selected variants that were marked as splice variants. These variants were then labeled as pathogenic if their clinical significance was annotated as pathogenic or likely pathogenic and benign if their clinical significance was annotated as benign or likely benign. This resulted in 35,464 variants labeled as pathogenic and 1001 labeled as benign. To calculate PR-AUC for delta scores we used 40,528 variants as negative examples, that had been determined to be highly unlikely to affect splicing based on differential splicing analysis in whole blood.

## Ethics and inclusion statement

This research received approval from the National Bioethics Committee of Iceland (approval number VSN 14-015) and was conducted in accordance with guidelines from the Icelandic Data Protection Authority (PV_2017060950þS/–). Informed consent was obtained from all participants, and an external party encrypted all personal identifiers before they were added to the deCODE database. All ethical regulations relevant to human research participants were followed.

Local researchers from deCODE genetics in Iceland were actively involved throughout the research process, including study design, implementation, and authorship of the publication. The research was developed in collaboration with local partners to ensure its relevance to the Icelandic population and the broader scientific community. The study did not involve any activities that are restricted or prohibited in the researchers' setting.

## Reporting summary

Further information on research design is available in the Nature Portfolio Reporting Summary linked to this article.

## Data availability

The data used in this study was generated from gene annotations obtained from ENSEMBL, GENCODE, and RNA-Seq data obtained through the GTEx Portal (https://gtexportal.org/home/datasets) and from an Icelandic cohort sequenced by deCODE genetics. ClinVar variants are available through https://ftp.ncbi.nlm.nih.gov/pub/clinvar/. The supplementary datasets, including model predictions and splice site annotations used in this study, are publicly available in the Zenodo repository at https://doi.org/10.5281/zenodo.14109868[46], and the underlying source data for Figs. 1–3 are available at https://doi.org/10.6084/m9.figshare.27607056[47]. The Icelandic RNA-Seq data used in this study are not publicly available due to information, contained within them, that could compromise research participant privacy, and releasing this information publicly is against Icelandic state law. Other data supporting the findings of this study are available from the corresponding authors upon reasonable request.

## Code availability

The code for preparing data, training models, and model evaluation, as well as the trained model weights, can be found in our GitHub repository https://github.com/benniatli/Spliceformer[48]. Usage examples for our pre-trained model are provided in a Google Colab notebook.

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

## Author contributions

B.A.J. implemented the method, wrote the code, and performed experiments. B.A.J., P.M., and M.O.U. developed the method and designed statistical experiments. G.H.H. and P.M. oversaw the processing RNA-Seq data and analysis of sQTLs in the Icelandic cohort. B.A.J., G.H.H., S.Á., S.R., E.E., P.S., D.F.G., P.M., K.S., and M.O.U. contributed to writing the final version of the manuscript.

## Competing interests

All authors are employed by deCODE Genetics/Amgen, Inc.
