## [Transparent Peer Review file · Communications Biology]

Transformers Significantly Improve Splice Site Prediction

Corresponding Author: Dr Magnús Úlfarsson

Version 0:

Reviewer comments:

Reviewer #1

(Remarks to the Author)

The authors present a new splice site prediction method, benchmark it to SpliceAI, and exploit the method to predict the consequences of variants present in the Icelandic population. The authors use SpliceAI to encode their data and add an attention and transformer step. The novelty is in the use of transformers with a large sequence context. The method does not demonstrate a large improvement in performance compared to SpliceAI. The number of sites that their model and SpliceAI disagree on is low and the impact on variant effect prediction in a research or clinical diagnostic setting can be disputed.

Major issues:

- The authors should prove that the improvements shown in Figure 1B, Table 1 and Table 2, are significant, as claimed in the Discussion. They train 10 different models but only show the average performance and no standard deviation. Including the standard deviation may support the suggested improved performance between their model and SpliceAI. Also in Table 1 and 2 the performance of the transformer and SpliceAI are similar, hence addition of standard deviations may help.
- The authors evaluate their model on sQTLs, but not on clinically relevant variants that were confirmed in the lab. Predicting the effect of variants in canonical splice motifs, as the authors do on page 7, is trivial. To make a bigger impact, the authors should show that their model detects more patient-relevant and experimentally confirmed splice sites that are the results of private variants.
- The authors should investigate whether the improvement in performance is due to the use of the transformer or due to the larger sequence context (45kb vs 10kb from SpliceAI-10k). The authors should add a comparison of the two methods using the same 10kb context for the transformer.
- It is unclear from the manuscript if both SpliceAI and the transformer model are finetuned or if only the transformer is finetuned. Moreover, the authors should specify how the finetuning was done. Were all weights retrained or were certain weights frozen?
- Transformers make use of attention and therefore are easier to interpret than convolutional neural networks. The authors only look at features like distance to nearest splice site in Figure S8 but it would also be interesting to see what motifs are created or disrupted by a variant.

Minor and textual issues:

- On page 3, "These sites are embedded" contains two times "are"
- Page 3: "select promising non-functional splice sites". Should be "select promising functional splice sites"??
- The legend of Figure 1a states that they look at 20kb surrounding the variant. Two times 20kb does not add up to the 45kb.
- In Figure 2 it is difficult to see the overlapping symbols for the Transformer and SpliceAI. Also, red and green are not good color choices for people suffering from color blindness.
- Discussion on page 8 "... since correct method ..." is not a correct sentence construction.

Reviewer #2

(Remarks to the Author)

The submitted manuscript presents a novel application of transformers to the problem of splice site prediction. This problem has been studied extensively over the past two decades and has witnessed an increased level of methods sophistication and a steady improvement in results. The paper by Jónsson et. al. presents continuation of this trend.

In general, the article is well written, the arguments are laid out clearly and a comprehensive supporting evidence of

incremental improvement relative to the current state-of-the-art methods is provided.

Whilst the article can be, in my opinion, published without changes, I would suggest that the authors consider addressing the following:

1. The starting premise of the method is that there is dependence among splice sites within the observed interval, thus having longer raw sequences clearly beneficial. Perhaps the authors could revisit the hypothesis in light of the results in the Discussion section. More specifically, would they consider the observed prediction improvements as validating the hypothesis to some extent?
2. The fine-tuning process on page 6 could be perhaps better elaborated.
3. Although all programs were provided in GitHub, an online interface to the predictor could simplify its use.

Reviewer #3

(Remarks to the Author)

This paper proposes a novel method utilizing Transformers for splice site prediction directly from raw 45 knt RNA sequences. The model leverages transcript annotations from GENCODE and ENSEMBL, alongside RNAseq cohort and GTEx data, to achieve improved prediction accuracy. The paper is well-structured and written, effectively conveying the methodology and supporting the conclusions with the presented results.

Major Comments:

1. The paper emphasizes the significance of non-canonical splice junctions. To enhance the accessibility of this valuable information, it would be highly beneficial to include a separate table or figure dedicated solely to non-canonical splice junctions. This would allow for easier retrieval and analysis of these data.
2. The authors highlight the use of one-hot encoding. Is there a specific motivation?
3. While the paper acknowledges potential time and resource limitations, it would be informative to understand the rationale behind employing only 10 epochs for the training process. Typically, achieving convergence in Transformer-based models often necessitates a significantly higher number of epochs.

Minor Comments:

4. The paper would benefit from providing additional context for the terms "hard/soft attention" and sQTLs for readers who may not be familiar with these concepts. This would ensure a more comprehensive understanding of the methodology employed.

Overall, the paper presents a promising approach using Transformers for splice site prediction. Addressing the major comments above would further strengthen the research's clarity and impact. Additionally, providing more context for the minor comments would enhance the paper's accessibility to a broader audience.

Version 1:

Reviewer comments:

Reviewer #1

(Remarks to the Author)

The authors have adequately responded to my comments. I would suggest to add the results from Transformer-10kb to the paper as suppl figure or suppl table.

Reviewer #3

(Remarks to the Author)

I am satisfied with the improvements the authors have made to the paper. The authors have successfully addressed several concerns raised in the previous review. Specifically, they have adequately addressed the concerns about the significance of the observed improvements over SpliceAI and conducted additional experiments to demonstrate that the method's performance is not solely due to the use of transformers or a larger sequence length.

Furthermore, the authors have provided more information on non-canonical splice junctions and justified their choice of one-hot encoding. They have also satisfactorily explained the number of epochs used for training the transformer. These changes have significantly strengthened the paper and addressed the major concerns raised in the previous review.

We want to thank the *Nature Communications Biology* referees for their insightful comments. We have carefully reviewed their feedback and addressed each point in detail below, incorporating the necessary revisions into our manuscript. The key points that have been addressed are concerns about the significance of the observed improvements over SpliceAI, that our method was not tested on clinically-relevant variants, and questions about whether the observed improvements were from using transformers or because the method has a larger sequence. To address the first point, we added 95% confidence intervals to the performance metrics displayed in Figure 1B, Table 1, and Table 2. These confidence intervals demonstrate that for both the area under the precision-recall curve (PR-AUC) and top-k accuracy, our method consistently outperforms SpliceAI, as the confidence intervals of our method were always above those of SpliceAI. To address the second point, we evaluated the classification performance on approximately 35,000 pathogenic splice variants from ClinVar [3]. Our findings show that our method achieved a higher PR-AUC than SpliceAI when tested on these clinically relevant variants. These results have been added to Figure 3C. To address the third point, we conducted an additional experiment where our model was trained using input sequences of the same size as those used by SpliceAI. The results indicate that even without increasing the context size, the transformer-based model outperforms SpliceAI, as shown in Figure 1B. We believe these revisions significantly enhance the robustness and clarity of our findings, and we appreciate the referees' contributions to improving our manuscript.

1 Reviewer 1

The authors should prove that the improvements shown in Figure 1B, Table 1 and Table 2, are significant, as claimed in the Discussion. They train 10 different models but only show the average performance and no standard deviation. Including the standard deviation may support the suggested improved performance between their model and SpliceAI. Also in Table 1 and 2 the performance of the transformer and SpliceAI are similar, hence addition of standard deviations may help.

- a. To examine the significance of the improvements shown in Figure 1B, Table 1, and Table 2, we have included 95% confidence intervals for all results. These confidence intervals demonstrate that our method consistently outperforms SpliceAI, as the intervals for our method are always above those for SpliceAI.

We opted to present confidence intervals rather than standard deviations because the results in Figure 1B, Table 1, and Table 2 are not derived from averaging performance metrics. Instead, we average the predictions for each nucleotide before calculating the performance metrics, resulting in a single test statistic for each metric. Consequently, standard deviations cannot be calculated from these single test statistics, making confidence intervals the appropriate measure for demonstrating the significance of our improvements.

We added a line to the Methods section, describing how these confidence intervals were calculated:

"To calculate 95% confidence intervals for PR-AUC and top-k accuracy we performed bootstrapping with 1000 samples."

The authors evaluate their model on sQTLs, but not on clinically relevant variants that were confirmed in the lab. Predicting the effect of variants in canonical splice motifs, as the authors do on page 7, is trivial. To make a bigger impact, the authors should show that their model detects more patient-relevant and experimentally confirmed splice sites that are the results of private variants.

- a. We agree that demonstrating our model's effectiveness at detecting clinically relevant splice variants is crucial. To address this, we have added classification results for pathogenic splice variants from ClinVar to Figure 3 [3]. This dataset includes 35,464 pathogenic splice variants and 40,528 control variants that are highly unlikely to affect splicing, as determined by differential splicing analysis in whole blood. Our results show that our method achieves a PR-AUC of 0.997, compared to SpliceAI's PR-AUC of 0.996.

It's important to explain why we included results for sQTLs. Predicting the effect of variants in canonical splice motifs might seem straightforward when accurate splice site annotations are available. However, methods like the one presented in our manuscript and SpliceAI do not rely on existing annotations to estimate the effects of variants. Instead, they aim to predict the annotations themselves, requiring highly accurate predicted splice sites to succeed. Therefore, even though predicting the impact of variants in canonical motifs might be simple with traditional approaches, it remains a challenging task for splice site prediction methods without the use of pre-existing annotations.

We have added a paragraph about the ClinVar variant classification to the Results section:

"We evaluated the classification performance of the Transformer-45k model on 35,464 pathogenic splice variants from ClinVar. The Transformer-45k achieved a PR-AUC of 0.997, compared to SpliceAI's PR-AUC of 0.996 (Figure 3c). To further analyze the predictive performance, we plotted a scatter plot of delta scores for these variants. The results indicate that pathogenic splice variants predominantly have delta scores around one, signifying high confidence in predictions. Non-splicing variants, on the other hand, cluster around a delta score of zero, indicating a low likelihood of splicing disruption. Interestingly, benign splice variants, though fewer in number (n = 1,001), exhibit a wider distribution of delta scores, which are less consistently clustered around zero and one (Figure 3d). This variability suggests a more complex prediction landscape for benign splice variants."

The authors should investigate whether the improvement in performance is due to the use of the transformer or due to the larger sequence context (45kb vs 10kb from SpliceAI-10k). The authors should add a comparison of the two methods using the same 10kb context for the transformer.

- a. Thank you for suggesting this experiment. We have trained a new version of the transformer model using a 10kb sequence context and added the results to Figure 1B. The performance metrics indicate that the transformer model with a 10kb context scores higher than SpliceAI, although it performs worse than the transformer model with a 45kb context. This suggests that the use of the transformer architecture itself contributes to more accurate predictions, even without increasing the context length. However, the larger sequence context still plays a significant role in enhancing the performance of these models.

The following text has been added to the Discussion:

"We conducted an additional experiment in Figure 1B to determine whether the observed performance improvement can be attributable to the transformer architecture or the increased sequence context. We trained a transformer model using a 10kb context and compared its performance to both SpliceAI and our 45kb context transformer model. The 10kb context transformer model outperformed SpliceAI, confirming that the transformer architecture contributes to more accurate predictions. Nevertheless, the 45kb context transformer model achieved the highest performance, highlighting that an extended sequence context is a significant factor in improving model accuracy."

It is unclear from the manuscript if both SpliceAI and the transformer model are finetuned or if only the transformer is finetuned. Moreover, the authors should specify how the finetuning was done. Were all weights re-trained or were certain weights frozen?

- a. Both the transformer model and SpliceAI were fine-tuned on RNA-Seq data. This allowed us to compare our model against a fine-tuned SpliceAI model, in addition to the pre-trained SpliceAI weights. We have added the following text to the Methods section to clarify how the fine-tuning was performed:

"To fine-tune the models on data from the Icelandic RNA-Seq cohort and GTEx V8, weights from the ENSEMBL dataset training runs were used as a starting point and trained for four additional epochs on splice sites obtained from RNA-Seq. During fine-tuning, all weights were kept trainable, and the learning rate was set to $2e-4$."

Additionally, the Processing of RNA-Seq data subsection in Methods includes information about how the splice site annotations, used for model fine-tuning, were created:

"The splice site annotations used for fine-tuning our model were constructed by combining RNA-Seq splice junctions detected in all 49 tissues in GTEx and the Icelandic blood samples. Junction reads were selected if they were present in four or more individuals, not in blacklisted sequencing regions, and if either end of the junction was present in the canonical transcript for a gene. The combined set of splice site annotations consists of 360,601 acceptors and 359,934 donors from 17,239 genes. In comparison, using the same method to construct annotations using exclusively reads from tissues in GTEx V8, we identified 310,532 acceptors and 311,499 donors from 16,308 genes."

Transformers make use of attention and therefore are easier to interpret than convolutional neural networks. The authors only look at features like distance to nearest splice site in Figure S8 but it would also be interesting to see what motifs are created or disrupted by a variant.

- a. Figure S8 displays sequence embeddings from *CFTR* extracted from the hg38 reference sequence. We cannot display any motifs that are created or disrupted by variants because there is no variation in the input.

Minor and textual issues:

- On page 3, "These sites are are embedded" contains two times "are"
 - Page 3: "select promising non-functional splice sites". Should be "select promising functional splice sites"??
 - The legend of Figure 1a states that they look at 20kb surrounding the variant. Two times 20kb does not add up to the 45kb.
 - In Figure 2 it is difficult to see the overlapping symbols for the Transformer and SpliceAI. Also, red and green are not good color choices for people suffering from color blindness.
 - Discussion on page 8 "... since correct method ... " is not a correct sentence construction.
- a. Thank you, we have corrected these typos and textual issues in the revised version of the manuscript. In Figure 2, we changed the color coding to orange and sky-blue to improve visibility for individuals with color blindness and adjusted the points by $\pm 20b$ to avoid overlapping symbols.

2 Reviewer 2

The starting premise of the method is that there is dependence among splice sites within the observed interval, thus having longer raw sequences clearly beneficial. Perhaps the authors could revisit the hypothesis in light of the results in the Discussion section. More specifically, would they consider the observed prediction improvements as validating the hypothesis to some extent?

- a. Dependence between splice sites, often referred to as competing splice sites, is a well-established concept in molecular biology [5]. This phenomenon occurs when multiple potential splice sites within a gene compete for recognition by the splicing machinery, influencing which sites are ultimately used and which exons are included or excluded in the final mRNA transcript. Such competition is influenced by various factors, including the intrinsic strength of the splice sites, the presence of splicing enhancers and silencers, and the binding of specific splicing factors [6].

One of the challenges faced by sequence-based splice site prediction models is the limited context window, which may be too short to capture the full context and dependencies between splice sites. This limitation can lead to inaccurate predictions, as the model might miss crucial regulatory elements that influence splice site selection.

In light of the improved predictions observed with longer sequence contexts, it seems plausible that extending the context window allows the model to better capture these dependencies, thereby validating the hypothesis to some extent. Specifically, by providing a broader sequence context, the model can learn the interactions between splice sites more effectively, leading to more accurate splicing predictions.

To address this point in the Discussion section, the following text has been added:

"By providing the transformer with a list of 512 potential splice sites, we enable it to produce more accurate predictions than those achieved with SpliceAI alone. This improvement may be attributed to the model's ability to learn the dependencies between splice sites over a larger sequence context, supporting the hypothesis that longer raw sequences are beneficial for capturing splice site interactions."

The fine-tuning process on page 6 could be perhaps better elaborated.

- a. We have provided a detailed explanation of the fine-tuning process in our response to Reviewer 1.

Although all programs were provided in GitHub, an online interface to the predictor could simplify its use.

- a. To simplify the use of our model, we have added a Google Colab notebook to the GitHub repository. This notebook allows users to generate predictions and delta scores for variants using pre-trained weights. The Google Colab notebook provides an accessible, interactive interface that does not require local setup or complex installations, making it easier for users to utilize our predictor. You can access the notebook here.

3 Reviewer 3

The paper emphasizes the significance of non-canonical splice junctions. To enhance the accessibility of this valuable information, it would be highly beneficial to include a separate table or figure dedicated solely to non-canonical splice junctions. This would allow for easier retrieval and analysis of these data.

- a. The non-canonical splice sites used for training are derived from RNA-Seq data, specifically from an Icelandic whole blood cohort and the GTEx project. An annotation table that includes the non-canonical splice sites can be generated using the code provided in our GitHub repository and publicly available GTEx data. However, to enhance the accessibility, we have added two supplementary tables that include splice site annotations derived from the Icelandic whole blood cohort and GTEx. Additionally, we have added the following text to the Data Availability section:

The splice site annotations used for fine-tuning and evaluating the models are included in Supplementary Table 1 (Icelandic whole blood combined with GTEx V8) and Supplementary Table 2 (GTEx V8 only).

The authors highlight the use of one-hot encoding. Is there a specific motivation?

- a. The inputs to the model are nucleotide sequences and since there are only four nucleotides it is natural to one hot encode them when feeding them to the model. This allows the model to work on finer resolution than if the inputs had been aggregated, such as is done in DNABERT and the Nucleotide Transformer [1, 7]. Other methods such as SpliceAI [2] and HyenaDNA [4] chose to not aggregate the sequence for the same reasons.

While the paper acknowledges potential time and resource limitations, it would be informative to understand the rationale behind employing only 10 epochs for the training process. Typically, achieving convergence in Transformer-based models often necessitates a significantly higher number of epochs.

- a. We limited the training process to 10 epochs for several reasons. First, we observed diminishing returns in the validation loss beyond this point, indicating that the model was beginning to converge. Additionally, we employed a learning rate scheduler that halved the learning rate after each epoch starting from the 5th epoch. This gradual reduction in the learning rate helped stabilize the training process and allowed the model to fine-tune its parameters effectively. While Transformer-based models typically require extensive data to achieve convergence, the substantial amount of data processed in each epoch of our dataset enabled the model to learn the necessary patterns within 10-epochs.

Minor Comments:

The paper would benefit from providing additional context for the terms "hard/soft attention" and sQTLs for readers who may not be familiar with these concepts. This would ensure a more comprehensive understanding of the methodology employed.

- a. We have added more information about "hard/soft attention" to the introduction:

"Hard attention selects a single or a few discrete elements from the input sequence by making a binary decision on which parts of the input to attend to, while soft attention, which is used in transformers, computes a weighted average over all elements of the input sequence. Using hard attention allows us to shorten the sequences that the transformer receives and to reduce the memory required to train the transformer."

And we have also added information about sQTLs to the introduction:

"Splicing quantitative trait loci (sQTLs) are a common class of variants that associate with usage of alternative splice sites in RNA."

References

- [1] Hugo Dalla-Torre, Liam Gonzalez, Javier Mendoza-Revilla, Nicolas Lopez Carranza, Adam Henryk Grzywaczewski, Francesco Oteri, Christian Dallago, Evan Trop, Hassan Sirelkhatim, Guillaume Richard, et al. The nucleotide transformer: Building and evaluating robust foundation models for human genomics. *bioRxiv*, pages 2023–01, 2023.
- [2] Kishore Jaganathan, Sofia Kyriazopoulou Panagiotopoulou, Jeremy F McRae, Siavash Fazel Darbandi, David Knowles, Yang I Li, Jack A Kosmicki, Juan Arbelaez, Wenwu Cui, Grace B Schwartz, et al. Predicting splicing from primary sequence with deep learning. *Cell*, 176(3):535–548, 2019.
- [3] Melissa J Landrum, Jennifer M Lee, George R Riley, Wonhee Jang, Wendy S Rubinstein, Deanna M Church, and Donna R Maglott. Clinvar: public archive of relationships among sequence variation and human phenotype. *Nucleic acids research*, 42(D1):D980–D985, 2014.
- [4] Eric Nguyen, Michael Poli, Marjan Faizi, Armin Thomas, Callum Birch-Sykes, Michael Wornow, Aman Patel, Clayton Rabideau, Stefano Massaroli, Yoshua Bengio, et al. Hyenadna: Long-range genomic sequence modeling at single nucleotide resolution. *arXiv preprint arXiv:2306.15794*, 2023.
- [5] Robin Reed and Tom Maniatis. A role for exon sequences and splice-site proximity in splice-site selection. *Cell*, 46(5):681–690, 1986.
- [6] Zefeng Wang and Christopher B Burge. Splicing regulation: from a parts list of regulatory elements to an integrated splicing code. *Rna*, 14(5):802–813, 2008.
- [7] Zhihan Zhou, Yanrong Ji, Weijian Li, Pratik Dutta, Ramana Davuluri, and Han Liu. Dnabert-2: Efficient foundation model and benchmark for multi-species genome. *arXiv preprint arXiv:2306.15006*, 2023.

1 Reviewer 1

The authors have adequately responded to my comments. I would suggest to add the results from Transformer-10kb to the paper as suppl figure or suppl table.

- a. Thank you for your suggestion. We have added the results from Transformer-10kb to Supplementary Table 3, complementing the results already presented in Figure 1b.

2 Reviewer 2

I am satisfied with the improvements the authors have made to the paper. The authors have successfully addressed several concerns raised in the previous review. Specifically, they have adequately addressed the concerns about the significance of the observed improvements over SpliceAI and conducted additional experiments to demonstrate that the method's performance is not solely due to the use of transformers or a larger sequence length.

Furthermore, the authors have provided more information on non-canonical splice junctions and justified their choice of one-hot encoding. They have also satisfactorily explained the number of epochs used for training the transformer. These changes have significantly strengthened the paper and addressed the major concerns raised in the previous review.

- a. Thank you for taking the time to review our manuscript again. We are delighted to hear that the revisions have satisfactorily addressed your concerns and strengthened the paper. Your insights have been helpful in improving the clarity and robustness of our work.